# Low Intensity Extracorporeal Shock Wave Therapy as a Potential Treatment for Overactive Bladder Syndrome

**DOI:** 10.3390/biology10060540

**Published:** 2021-06-16

**Authors:** Jian-He Lu, Kuang-Shun Chueh, Shu-Mien Chuang, Yi-Hsuan Wu, Kun-Ling Lin, Cheng-Yu Long, Yung-Chin Lee, Mei-Chen Shen, Ting-Wei Sun, Yung-Shun Juan

**Affiliations:** 1Emerging Compounds Research Center, Department of Environmental Science and Engineering, College of Engineering, National Pingtung University of Science and Technology, Pingtung County 91201, Taiwan; toddherpuma@yahoo.com.tw; 2Department of Urology, College of Medicine, Kaohsiung Medical University, Kaohsiung 80708, Taiwan; spacejason69@yahoo.com.tw (K.-S.C.); u9181002@gmail.com (S.-M.C.); leeyc12345@yahoo.com.tw (Y.-C.L.); bear5824@gmail.com (M.-C.S.); selina750220@yahoo.com.tw (T.-W.S.); 3Department of Urology, Kaohsiung Medical University Hospital, Kaohsiung 80756, Taiwan; maivy0314@gmail.com; 4Graduate Institute of Clinical Medicine, College of Medicine, Kaohsiung Medical University, Kaohsiung 80708, Taiwan; nancylin95@gmail.com (K.-L.L.); urolong@yahoo.com.tw (C.-Y.L.); 5Department of Urology, Kaohsiung Municipal Ta-Tung Hospital, Kaohsiung 80661, Taiwan; 6Department of Obstetrics and Gynecology, Kaohsiung Municipal Ta-Tung Hospital, Kaohsiung 80661, Taiwan; 7Department of Obstetrics and Gynecology, Kaohsiung Medical University Hospital, Kaohsiung 80756, Taiwan; 8Department of Obstetrics and Gynecology, Kaohsiung Municipal Hsiao-Kang Hospital, Kaohsiung 81267, Taiwan; 9Regenerative Medicine and Cell Therapy Research Center (RCC), Kaohsiung Medical University, Kaohsiung 80708, Taiwan; 10Department of Urology, Kaohsiung Municipal Hsiao-Kang Hospital, Kaohsiung 81267, Taiwan

**Keywords:** overactive bladder, low intensity extracorporeal shock wave therapy, urinary frequency and urgency

## Abstract

**Simple Summary:**

Overactive bladder (OAB) is a common urologic condition with urinary frequency, urinary urgency, nocturia, and urgency incontinence, which can get in the way of a patient’s social life, exercise, work, and sleep. Exploring a promising option for OAB patients is very important, especially one with less side effects or invasive alternations. This study uses low intensity extracorporeal shock wave therapy (LiESWT) to investigate the therapeutic effect and duration on OAB symptoms.

**Abstract:**

Background: The present study attempted to investigate the therapeutic effect and duration of low intensity extracorporeal shock wave therapy (LiESWT) on overactive bladder (OAB) symptoms, including social activity and the quality of life (QoL). Methods: In this prospective, randomized, single-blinded clinical trial, 65 participants with OAB symptom were randomly divided into receive LiESWT (0.25 mJ/mm^2^, 3000 pulses, 3 pulses/second) once a week for 8 weeks, or an identical sham LiESWT treatment without the energy transmission. We analyzed the difference in overactive bladder symptom score (OABSS) and 3-day urinary diary as the primary end. The secondary endpoint consisted of the change in uroflowmetry, post-voided residual (PVR) urine, and validated standardized questionnaires at the baseline (W0), 4-week (W4) and 8-week (W8) of LiESWT, and 1-month (F1), 3-month (F3) and 6-month (F6) follow-up after LiESWT. Results: 8-week LiESWT could significantly decrease urinary frequency, nocturia, urgency, and PVR volume, but meaningfully increase functional bladder capacity, average voided volume and maximal flow rate (Qmax) as compared with the W0 in the LiESWT group. In addition, the scores calculated from questionnaires were meaningfully reduced at W4, W8, F1, F3, and F6 in the LiESWT group. Conclusions: Our results revealed that the therapeutic efficacy of LiESWT could improve voided volume and ameliorate OAB symptoms, such as urgency, frequency, nocturia, and urinary incontinence, and lasted up to 6 month of follow-up. Moreover, LiESWT treatment brought statistically significant and clinically meaningful improvements in social activity and QoL of patients. These findings suggested that LiESWT could serve as an alternative non-invasive therapy for OAB patients.

## 1. Introduction

Four characteristically symptoms of overactive bladder (OAB) bothered patients a lot and severely destroyed the quality of life, including urinary frequency, urinary urgency, nocturia, and urgency incontinence, were found in approximately 20.9% of females, and is increased to 34.5% in those who were over 65 year old in Asian countries [1,2,3]. The pathophysiology of OAB is composed of multiple possible causes which are not fully elucidated. Its mechanism may be related to bladder innervation, purinergic and muscarinic receptors, and abnormal increase in the production of prostaglandin and leukotriene [4,5]. Some studies suggested that oxidative stress, hypoxia, and decreased blood supply may play an important role [6,7]. In terms of expenses for the healthcare, OAB symptoms indeed placed a large burden on these patients. 

Clinicians first educated patients about lifestyle-modification and behavior adjustment. Following conservative managements, anti-muscarinic agents, and/or ß3 agonist medication were composed of main pharmacotherapy in OAB. Nevertheless, patients were bothered by constipation, dry mouth, and blurred vision, especially after using anti-muscarinic agents. Besides, severe adverse effects, such as retention of urine, urinary tract infection (UTI) [8], and cognitive change [9] were significantly higher in the aging population. Hence, only 8.3–24% of patients maintained long-term usage [10]. Several side effects, including somnolence, dizziness, and changes in blood pressure may occur after using ß3 agonist; this was the reason why the continuous rate after 12 months dropped to 38% [10]. Intravesical botulinum toxin A injection, percutaneous tibial nerve stimulation, and sacral neuromodulation were thought to be the third-line treatment once all non-invasive managements failed [11,12]. The botulinum toxin A inhibited the pre-synaptic release of acetylcholine, and subsequently prevented detrusor muscle contraction which was triggered by parasympathetic efferent nerve. Nevertheless, overly altering the afferent nerve input and inhibiting detrusor smooth muscle contraction may result in considerable post-voided residual urine (PVR) (>150 mL), acute urinary retention, and UTI [11]. Moreover, such side effects may dramatically lower the patients’ compliance and persistent rate of medical treatment. Hence, it is the most important mission for us to explore a promising option for OAB patients, especially with less side effect or invasive alternation. 

Extrapolating from previous studies, low intensity extracorporeal shock wave therapy (LiESWT) has been used widely for various types of urological diseases. For example, LiESWT treatment (0.10–0.25 mJ/mm^2^ and 3000–6000 pulses for 4–8 weeks) increased penile hemodynamics and induced penile tissue regeneration in erectile dysfunction (ED) patients [13,14,15]. LiESWT (0.10 to 0.25 mJ/mm^2^, 3000 impulses for 4 weeks) was found to significantly improved pain, micturition, erectile function, and quality of life (QoL) in chronic pelvic pain syndrome (CPPS) patients. It might be used as a potential therapy in treating non-bacterial prostatitis/CPPS [16]. We discovered a pleasant result for severe CPPS patients who were refractory to many kinds of medical therapy, the pain scale and the severity of urinary symptoms were significantly improved after the LiESWT [17]. Additionally, our previous study suggested LiESWT for stress urinary incontinence (SUI) patients attenuated bladder leaks and improved QoL and social activity after 8-week LiESWT (0.25 mJ/mm^2^ and 3000 pulses) [18]. 

In the present research, we hypothesized LiESWT could reduce bladder inflammation, increase angiogenesis, ameliorate OAB symptoms, and thus improve social activity and QoL. The goal of the present study is to evaluate the therapeutic efficacy and duration of LiESWT on OAB participants, including ameliorating OAB symptoms, increasing voided volume, and improving patients’ social activity and QoL.

## 2. Materials and Methods

### 2.1. Design

The current prospective, randomized, single-blinded trial was performed in a tertiary medical center from December 2018 through January 2020. It was carried out under the approval of Institutional Review Board (IRB No. KMUHIRB-F(II)-20180010) and was registered at clinicaltrials.gov (NCT04059133) on 16 August 2019. The major inclusion and exclusion criteria are shown in Table 1. 65 female eligible participants recruited from outpatient clinics were provided with informed consent before entering the study and were randomly allocated to the sham group (n = 15) or the LiESWT group (n = 50) by computer generated random numbers. The timetable designed for clinical trial of OAB is shown in Figure 1.

### 2.2. Procedure and Setting of LiESWT

The LiESWT device we used was the DUOLITH SD1-TOP focused shock wave system (STORZ MEDICAL, AG, Kreuzlingen, Switzerland). Before LiESWT procedure, the functional bladder capacity was determined by 3-day urinary diary data and uroflowmetry data. All participants were asked to drink 1000 mL of water, then received LiESWT treatment after filling the bladder to 50% of the functional bladder capacity by using bladder scan sonography. An experienced urologist gently placed the probe on the patient’s lower abdomen around two fingers above the pubic symphysis. The probe of the LiESWT group with silicone pad was tilted to 45° to target the bladder dome and bilateral walls (each side 1000 impulses, total 3000 pulses per treatment session) at 0.25 mJ/mm^2^ of energy, and 3 pulses/second of frequency [18,19], once a week for 8 weeks. The sham (placebo) group used identical treatment. The probe of sham (placebo) group used air pad to block energy transmission, but the machine still emitted shock wave generation. 

### 2.3. Physical and Serum Biochemical Indicators of Studied Participants

Metabolic syndrome was associated with the symptoms of OAB, so we retrieved the related indicators to analyze baseline condition of OAB population. Physical indicators (age, height, weight, waistline, body mass index, and blood pressure) and serum biochemical parameters (glycated hemoglobin A1C, fasting blood sugar), renal function index (blood urea nitrogen, creatinine), liver function index (glutamate oxaloacetate transaminase and glutamate pyruvate transaminase), and lipid profile (triglycerides, cholesterol, low-density lipoprotein, and high-density lipoprotein)] were collected [20].

### 2.4. Outcome Measures and Therapeutic Efficacy Assessment for LiESWT

To assess the therapeutic efficacy of LiESWT, the primary endpoint was designated as the change in Overactive Bladder Symptom Scores (OABSS) and 3-day urinary diary [18]. The secondary endpoint was the change in uroflowmetry (voided urine volume and maximal flow rate [Qmax]), PVR as well as life bothersome questionnaires, including International Consultation on Incontinence Questionnaire-Short Form (ICIQ-SF), Urogenital Distress Inventory-6 (UDI-6) and Incontinence Impact Questionnaire-7 (IIQ-7) [18] at the baseline (W0), 4-week (W4), 8-week (W8), 1-month follow-up (F1), 3-month follow-up (F3) and 6-month follow-up (F6). 

### 2.5. Statistical Analysis

Statistical analyses were performed using SAS 9.3 (SAS Institute, Cary, NC, USA). Quantitative data were represented as the mean ± standard error (SE). Student *t*-test was performed for the intergroup comparison, and the paired *t*-test and one-way analysis of variance were used to perform repeated measurement analyses for intragroup before/after treatment [21]. In order to clarify the therapeutic effect of LiESWT on OAB, the scores of pre- and post-treatment for intragroup of patients were compared. The paired *t*-test was performed in the sham group (W0 vs. W4). The post-hoc Tukey’s honestly significant difference tests were used to make comparison between the LiESWT subgroups and to calculate *p*-values for comparison [21]. On the other hand, the intergroup relationship (sham group vs. LiESWT group) of W0 and W4 data were evaluated by using the student’s *t*-test. In these analyses, *p* < 0.05 represented statistically significant. 

## 3. Results

### 3.1. Diagnoses

Physical and biochemical parameters of the OAB participants were analyzed in Table 2. The mean age of the sham group was 55.63 ± 2.07 years, while that in the LiESWT group was 57.01 ± 1.14 years, respectively. No significant difference in baseline parameters was found between the two groups. 

### 3.2. Primary and Secondary End Points

All of the participants suffered from daytime frequency ≥8 times (100%) and nocturia ≥1 time (100%). Besides, the proportion of urgency ≥1 time and urgency incontinence ≥1 time were 90% and 73% in the sham group as well as 90% and 63% in the LiESWT group, respectively (Figure 2a). In comparison with the baseline, there was no meaningful difference in the daytime frequency (*p* = 0.77), nocturia (*p* = 0.12) and urgency (*p* = 0.50) after 4 weeks of treatment in the sham group. However, 4-week LiESWT significantly improved daytime frequency (*p* < 0.01) of OAB symptoms, as compared with W0 (Figure 2b). Besides, LiESWT meaningfully decreased daytime frequency (*p* < 0.01), nocturia (*p* < 0.05) and urgency symptoms (*p* < 0.05) at W8, F1, F3, and F6, as compared with the W0 (Figure 2c). Taken together, these findings revealed that LiESWT improved the OAB symptoms by decreasing daytime frequency as early as 4 weeks and this effect was more prominent after 8 weeks treatment on daytime frequency, nocturia, and urgency. Moreover, such a therapeutic effect could last until after 6 months of follow-up.

To further research the therapeutic effect and duration of LiEWST in OAB patients, we analyzed the change of 3-day urinary diary, presented in Table 3. In comparison with W0, there was no significant difference in the amount of fluid intake, urine output, average voided volume, functional bladder capacity, daytime frequency, nocturia, or urgency time after 4 weeks treatment in the sham group. However, 4-week LiESWT significantly decreased daytime frequency compared to the sham group ((*p* = 0.03). Moreover, the mean values of average voided volume (mL) were noticeably increased from W0 (186.9 ± 8.4) to W8 (224.6 ± 8.6) (*p* = 0.02 in the LiESWT group. The mean values of functional bladder capacity (mL) were obviously raised from W0 (330.5 ± 12.2) to W8 (375.5 ± 13.2) (*p* < 0.05) in the LiESWT group (Table 3).

With regard to OAB symptoms, LiESWT meaningfully decreased the mean values of daytime frequency (11.85 ± 0.50 vs. 9.48 ± 0.33, *p* < 0.01), nocturia (1.77 ± 0.17 vs. 1.20 ± 0.12, *p* < 0.01), and urgency (3.16 ± 0.39 vs. 1.96 ± 0.34, *p* = 0.04) from W0 to W8, respectively. Such effects were sustained until F6 (Table 3). These data revealed that LiESWT significantly ameliorated OAB symptoms and improved bladder function at W8, F1, F3, and F6, as compared with W0.

The bladder voiding function was objectively determined by uroflowmetry (voided urine volume and Qmax) and PVR (Table 3). In comparison with W0, there was no significant difference in the voided urine volume, Qmax or PVR in the sham group. However, the mean value of voided urine volume (mL) was significantly expanded from 296.2 ± 16.7 (W0) to 357.5 ± 20.1 (W4, *p* < 0.05). The mean of Qmax (mL/s) was significantly increased from 25.61 ± 1.69 (W0) to 30.51 ± 1.26 (W8, *p* < 0.05). Predictably, PVR (mL) visibly reduced from W0 (49.63 ± 6.18) to F1 (30.04 ± 3.90, *p* < 0.05). All of them were noticeably improved at W8, F1, F3, and F6 as compared with W0 and those in the sham group. The present results demonstrated that 8-week LiESWT caused significant improvements in voiding efficacy. Such beneficial effects could last as long as sixth months after follow-up.

We investigate the efficacy of LiWEST on the social activity and the QoL based on a series of well-known questionnaires, including OABSS, ICIQ-SF, UDI-6, and IIQ-7 (Figure 3). In comparison with W0 data, there was no meaningful difference in the scores of OABSS (*p* = 0.18), ICIQ-SF (*p* = 0.37), UDI-6 (*p* = 0.99) and IIQ-7 (*p* = 0.12) in the sham group (Figure 3a). However, in the 4-week LiESWT group, the scores of OABSS (*p* < 0.01), UDI-6 (*p* < 0.01) and IIQ-7 (*p* < 0.01) had significantly declined as compared with W0. Moreover, 4-week LiESWT also noticeably decreased OABSS (*p* = 0.04), UDI-6 (*p* = 0.04) and IIQ-7 (*p* = 0.02) compared to the sham group. According to questionnaire scores, the scores of OABSS, ICIQ-SF, UDI-6 and IIQ-7 were significantly decreased at W4, W8, F1, F3, and F6 compared to W0 (Figure 3c). These findings revealed that LiWEST indeed make symptoms relief and enhance the QoL in the OAB population.

According to the subgroups of OABSS (Table 3 and Figure 3b), there was no significant difference in the daytime frequency (*p* = 0.99), nocturia (*p* = 0.25), urgency (*p* = 0.43) or urgency incontinence (*p* = 0.75) of the sham group, as compared with W0 (Figure 3b). However, 4-week LiESWT noticeably decreased daytime frequency (*p* = 0.04), nocturia (*p* = 0.04), urgency (*p* = 0.04) or urgency incontinence (*p* = 0.04) compared to the sham group (Figure 3b). With regard to OAB symptoms, the mean times of daytime frequency reduced from W0 (1.22 ± 0.07) to W4 (0.84 ± 0.04, *p* < 0.01) in Table 3 and Figure 3d. Nocturia significantly decreased from W0 (2.30 ± 0.10) to W4 (1.45 ± 0.11, *p* < 0.01). The mean time of the urgency decreased from W0 (2.86 ± 0.19) to W4 (1.70 ± 0.15, *p* < 0.01). Furthermore, the mean time of the urgency incontinence decreased from W0 (1.94 ± 0.21) to W4 (0.89 ± 0.14, *p* < 0.01). Such effects were sustained until 6-month follow-up (Table 3, Figure 3b,d). 

### 3.3. Safety of LiESWT Treatments

As far as safety was concerned, LiESWT treatment was well tolerated by all of participants. There were no known adverse effects, such as intolerable pain, gross hematuria, or skin ecchymosis, in association with LiESWT in this study.

### 3.4. A Brief Diagram Proposed for the Potential Effects of LiESWT

Basing on the above findings, a brief diagram was presented for the therapeutic effects of LiESWT for OAB participants (Figure 4). OAB symptoms included daytime frequency, nocturia, urgency or/and urgency incontinence. The effects of LiESWT (3000 pulses at 0.25 mJ/mm^2^ of energy and 3 pulses/second of frequency) on OAB were analyzed using 3-day urinary diary, uroflowmetry, PVR, and questionnaire scores. Here we demonstrated that LiESWT improved bladder function and ameliorated the OAB symptoms, after 8 weeks of LiESWT treatment. In addition, such effects greatly promoted participants’ social activity and QoL.

## 4. Discussion

Chronic OAB syndrome composed of annoying symptoms caused tremendous adverse impact on the QoL. Urinary frequency, urgency, nocturia, and urinary incontinence substantially and psychosocially destroyed patients’ joy of life and esteem. In the present study, 3-day urinary diary, OABSS evaluation, uroflowmetry, PVR, and life bothersome questionnaires were analyzed. The results revealed that LiESWT not only improved the functional bladder capacity and the average voided volume, but also reduced OAB symptoms. Additionally, the LiESWT decreased the amount of PVR, indicating improvement in voiding efficacy. It took eight weeks to notice such favorable effects after receiving LiESWT. The present findings suggested that LiESWT could serve as a potential non-medical, non-invasive therapy for OAB.

The overall prevalence of OAB in adults was around 11.8%, while there was no significant gender difference in the incidence [22]. Although the prevalence rates in both genders were similar, sex specific differences such as the distribution of neurotransmitter expression and its sensitivity to hormone fluctuations do exist in relation to individual OAB symptoms as well as the degree of bother, therapeutic response, and the QoL [23]. LiESWT has been applied in men with ED [13,14,15,24] or with CPPS [16,17,25,26,27], and in female with SUI [18]. However, more trials with different protocols are required before LiESWT can be applied in OAB patients. The present study revealed that LiESWT not only ameliorated OAB symptoms, but also improved bladder voiding function in female participants. As is well known, the incidence of OAB increased in the aging population and OAB was correlated with the estrogen deficiency. In the present study, half of participants with OAB symptoms were menopausal. It would be interesting to explore the therapeutic efficacy and the molecular mechanism of LiESWT on menopausal OAB subjects in a future study.

Current evidence unveiled that LiESWT ameliorated OAB symptoms and improved bladder function and QoL as well, though the underlying pathophysiology was unclear. It is difficult and ethically controversial to obtain human bladder tissue. Instead, it became necessary to be devoted to molecular mechanism of LiESWT through fundamental laboratory experiment or animal models. In rat models, LiESWT improved OAB and bladder functions through inducing promoted angiogenesis, decreased inflammation reaction by reducing oxidative stress [28,29], ameliorated diabetic bladder dysfunction and urinary incontinence [30]. Meanwhile, the effect of LiESWT also increased bladder nerve innervation and vascularization, enhanced bladder and urethra muscle contractile function, activated bladder muscle regeneration, ameliorated bladder wall composition, and enhanced urethra continence mechanism [30,31]. It is a kind of non-invasive novel therapy with promising effects for non-bacterial prostatitis/CPPS. We proposed therapeutic mechanisms of LiESWT included neovascularization and inflammatory reduction by enhancing expression of vascular endothelial growth factor (VEGF), endothelial nitric oxide synthase (eNOS), and proliferative cell nuclear antigen (PCNA), ultimately promoting regeneration of bladder tissue. Therefore, we simultaneously investigated the molecular mechanism of LiESWT on OAB through rat experiments.

The molecular mechanism of LiESWT activated protein kinase RNA-like ER kinase pathway by increasing the phosphorylation levels of PERK and eukaryotic initiation factor 2a (eIF2α) [32] and intensifying activating transcription factor 4 (ATF4) to increase the myotube formation in rat myoblast cells [33]. Zhu et al. showed the effect of the combination of mesenchymal stem cells (MSC) and LiESWT induced vasodilatation and angiogenesis on PI3K/AKT/mTOR signal pathway and activated NO/cGMP signal pathway [34]. Additionally, LiESWT has protective effects on inflammation through lowering the expression of NGF, IL-6, IL-12, TNF-α, COX-2 and iNOS [29,35]. LiESWT could enhance endothelial NO synthase (eNOS) activity, which would result in the suppression of NF-κB to decrease inflammation [36].

Considering participant safety and interference factors, we have excluded patient with severe coagulopathy, urologic cancer, previous pelvic radiation, and significant bladder outlet obstruction. Although LiESWT has little adverse effect in general practice, severe coagulopathy and malignant tumor at treatment sites were still considered as absolute contraindications. We also excluded pregnant women because fetus in the treatment area is another contraindication. Abdominal fat distribution, including subcutaneous fat and visceral fat may influence the depth of the treatment site. Fortunately, most participants in the study are thin and have a BMI between 18.6–26, abdominal fat accumulation has little effect on the present study. In order to improve the effectiveness of LiESWT, the probe (applicator) was placed on the patient’s lower abdomen around two fingers above the pubic symphysis. The probe was tilted to 45° to target the bladder dome and bilateral bladder walls. Therefore, the effect of abdominal fat thickness will be decrease to the least level. The usage of ultrasound transmission gel over the abdominal skin and applicator to avoid air interference is also very important. All participants were asked to drink water, then received LiESWT treatment after filling the bladder to half of the functional bladder capacity by using bladder scan sonography. During LiESWT treatment, some participants felt a slight tingling sensation, while others experienced no sensation. Overall, there were no adverse effects, such as intolerable pain, gross hematuria, or skin ecchymosis experienced by the participants during the shock wave treatment.

A statistically significant difference is not equivalent to a meaningful clinical change or meaningful improvement in patient bother. However, Gotoh et al. had revealed a linear tendency between the changes in the OABSS and symptom improvement. They also showed a change -3 in OABSS total score is the minimal threshold for a clinical meaningful change [37]. In the present study (Table 3), in the sham group, the OABSS scores only decreased from the baseline W0 (7.80 ± 0.68) to W4 (6.20 ± 0.98). However, in the treatment group, the OABSS scores decreased from the baseline W0 (7.42 ± 0.41) to W8 (3.75 ± 0.25), F1 (3.23 ± 0.20), F3 (3.38 ± 0.24), and F6 (3.50 ± 0.26), respectively. The difference in OABSS between pre-treatment and post-treatment is greater than 3, showing that LiESWT not only had statistically significant difference but also provided clinically meaningful improvement for these OAB patients.

There were several limitations in the present study. First, some participants of the sham group withdraw at the fifth or the sixth course because there was no obvious improvement after sham treatment without energy transmission. The data in the sham group were only W4 data and lack of W8, F1, F3, and F6. Therefore, the number of participants in the sham group (15 subjects) was different from the LiESWT group (50 subjects). Second, we included only female participants. The benefits of LiESWT on male participants would be an interesting project for our future exploration. Third, to determine the optimal protocol for OAB patients, different pulse intervals and frequencies should be evaluated in future experiments. 

## 5. Conclusions

The present study revealed that LiESWT improved the OAB symptoms, enhanced voiding efficacy, and promoted QoL of OAB participants. These findings suggested that LiESWT may be a potential non-medical and non-invasive alternative therapy for OAB patients.

## Figures and Tables

**Figure 1 biology-10-00540-f001:**
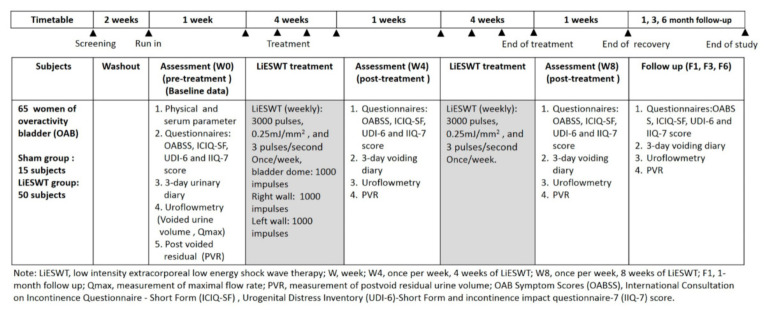
Timetable designed for clinical trial of overactive bladder (OAB) during LiESWT treatment procedure. LiESWT: low intensity extracorporeal shock wave therapy, Qmax: measurement of maximal flow rate, PVR: measurement of post-voided residual (PVR) urine volume, OABSS: overactive bladder symptom scores, ICIQ-SF: international consultation on incontinence questionnaire-short form, UDI-6: urogenital distress inventory-short form, IIQ-7: incontinence impact questionnaire-7.

**Figure 2 biology-10-00540-f002:**
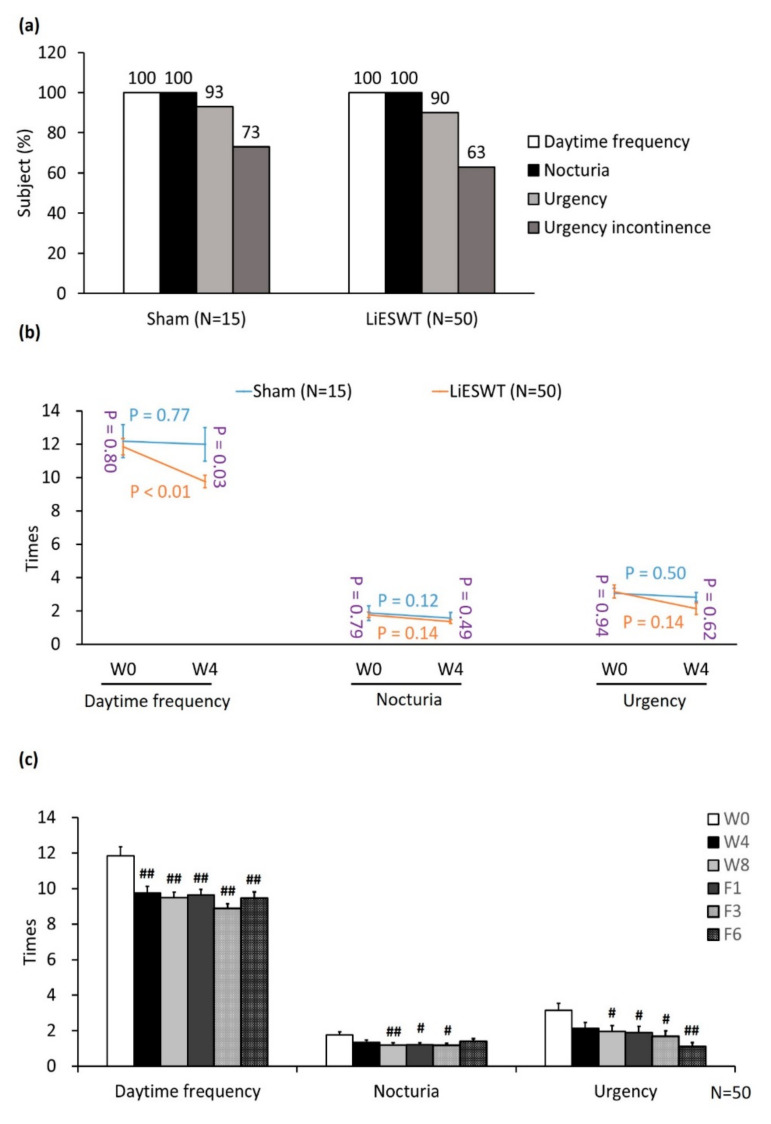
Analysis of studied population with OAB symptoms. (**a**) The percentage of studied population with OAB symptoms. OAB symptoms consisted of daytime frequency, nocturia, and urgency with or without urgency incontinence. (**b**) The changes in daytime frequency, nocturia and urgency at W4, as compared with W0. The blue or orange font denotes the *p*-value before and after 4 weeks treatment in the sham group or in the LiESWT-treated group, respectively. The purple font indicated the *p*-value between the sham group and the LiESWT-treated group at W0 and W4. (**c**) LiESWT improved the OAB symptoms. The mean values of daytime frequency, nocturia and urgency were meaningfully decreased after LiESWT treatment at W8. W0: the baseline, W4: 4-week of LiESWT treatment, W8: 8-week of LiESWT treatment, F1: 1-month follow-up, F3: 3-month follow-up, F6: 6-month follow-up. ^#^
*p* < 0.05; ^##^
*p* < 0.01 as compared with W0.

**Figure 3 biology-10-00540-f003:**
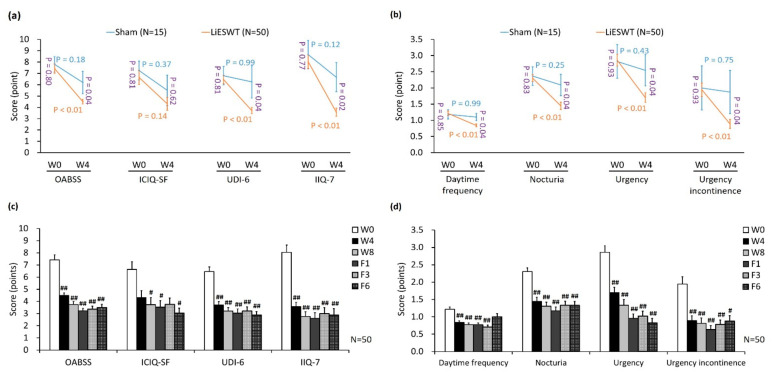
Improvement of OAB symptoms and life bothersome questionnaire scores after LiESWT treatment. (**a**,**c**) The therapeutic effect of LiESWT was analyzed by the OAB symptoms and life bothersome questionnaires, including OABSS, ICIQ-SF, UDI-6, and IIQ-7. LiESWT treatment significantly reduced the scores of OABSS, ICIQ-SF, UDI-6, and IIQ-7 as compared with the sham group. (**b**,**d**) Improvement of questionnaires scores for OAB symptom after LiESWT treatment, including daytime frequency, nocturia, urgency, and urgency incontinence. LiESWT improved OAB symptoms and the QoL. OABSS, overactive bladder symptom scores. ICIQ-SF: international consultation on incontinence questionnaire-short form, UDI-6: urogenital distress inventory–6, IIQ-7: incontinence impact questionnaire-7, W0: the baseline, W4: 4-week LiESWT treatment, W8: 8-week LiESWT treatment, F1: 1-month follow-up, F3: 3-month follow-up. The blue or orange font denotes the *p*-value before and after 4 weeks treatment in the sham group or in the LiESWT group, respectively. The purple font indicates the *p*-value between the sham group and the LiESWT group at the W0 and W4. ^#^
*p* < 0.05; ^##^
*p* < 0.01 compared to W0.

**Figure 4 biology-10-00540-f004:**
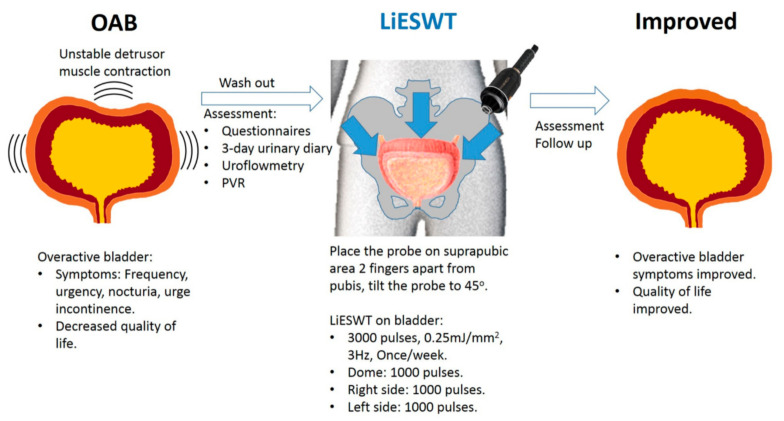
A brief diagram proposed for the potential effects of LiESWT on improving the OAB symptoms. OAB: overactive bladder, LiESWT: low intensity extracorporeal shock wave therapy, PVR: measurement of post-voided residual urine volume.

**Table 1 biology-10-00540-t001:** Inclusion and exclusion criteria.

Inclusion Criteria	Exclusion Criteria
1. Female participants aged 20–75 years who were diagnosed with OAB for more than 3 months. 2. OAB symptoms included daytime frequency of micturition ≥ 8 times, and nocturia, urgency or urgency incontinence ≥ 1 times. 3. Patients could understand and follow the instructions and were able to complete the questionnaire. 4. Patients with OAB symptom who did not take antimuscarinic or ß3 agonist. 5. OAB patient with antimuscarinic or ß3 agonist treatment could also be included after 3 months of medication withdrawal 6. Signature of informed consent form.	1. Urinary tract infection detected at screening, and recurrent for urinary tract infections more than 3 episodes in the past 3 months. 2. Neuropathic diseases. 3. Lower urinary tract surgery within last 6 months. 4. Significant bladder outflow obstruction. 5. Drug or nondrug treatments of OAB in the previous 3 months. 6. History of urolithiasis or urologic cancer. 7. Gross hematuria. 8. Severe cardiopulmonary disease, coagulopathy, liver or renal dysfunction. 9. Previous pelvic radiation therapy. 10. Women with pregnancy or within six months of childbearing.

Note: OAB, overactive bladder.

**Table 2 biology-10-00540-t002:** Baseline characteristics of overactive bladder (OAB) population.

Parameter	Sham (Mean ± SE)	LiESWT (Mean ± SE)	Normal Range
Physical parameter			
Female age (years)	55.63 ± 2.07	57.01 ± 1.14	20–75
Height (cm)	160.68 ± 1.15	157.07 ± 0.57	
Weight (kg)	64.81 ± 3.66	58.24 ± 0.95	
BMI (kg/m^2^)	24.99 ± 1.19	24.39 ± 0.80	18.5–26
Waistline (cm)	88.69 ± 3.20	85.07 ± 1.07	
Systolic pressure (mmHg)	115.50 ± 3.85	119.32 ± 1.50	100–120
Diastolic pressure (mmHg)	66.88 ± 2.06	72.08 ± 1.03	60–80
MAP (mmHg)	83.08 ± 2.41	87.83 ± 1.04	70–110
Serum parameter			
HbA1C (%)	5.75 ± 0.11	5.61 ± 0.08	
AC sugar (mg/dl)	99.75 ± 1.32	101.11 ± 2.69	65–109
BUN (mg/dl)	10.98 ± 0.89	12.50 ± 0.50	8–20
Creatinine (mg/dl)	0.70 ± 0.04	0.72 ± 0.02	0.44–1.03
GOT(AST) (IU/L)	22.25 ± 1.07	25.56 ± 1.66	10–42
GPT(ALT) (IU/L)	24.63 ± 2.33	22.88 ± 1.60	10–40
Triglycerides (mg/dl)	88.63 ± 5.79	90.30 ± 5.12	35–160
Cholesterol (mg/dl)	215.50 ± 9.84	206.54 ± 4.57	140–200
HDL (mg/dl)	63.88 ± 3.32	58.38 ± 1.88	29–85
LDL (mg/dl)	126.41 ± 6.84	122.54 ± 4.11	0–130

Note: BMI, body mass index; MAP, mean arterial pressure; HbA1C, hemoglobin A1c (glycated hemoglobin); AC, Ante Cibum (before meals); BUN, blood urea nitrogen; GOT, glutamate oxaloacetate transaminase; GPT, glutamate pyruvate transaminase; LDL, low-density lipoprotein; HDL, high-density lipoprotein; Values are means ± SE. N = 15 (Sham) and N = 50 (LiESWT).

**Table 3 biology-10-00540-t003:** Urodynamic parameters of study population for overactive bladder (OAB).

Parameter	Sham (N = 15)		LiESWT (N = 50)					
	W0	W4	W0	W4	W8	F1	F3	F6
3-day urinary diary record								
Intake (mL)	2094.8 ± 208.7	2196.5 ± 111.2	1987.0 ± 85.0	1906.8 ± 82.3	2180.9 ± 221.5	1921.3 ± 93.0	1969.3 ± 97.7	1867.0 ± 55.7
Output (mL)	2321.9 ± 250.4	2219.0 ± 166.6	2113.4 ± 89.5	2017.3 ± 79.4	2007.7 ± 83.4	2064.2 ± 82.0	2034.0 ± 93.4	1878.2 ± 64.4
Average voided volume (mL)	196.4 ± 10.8	211.8 ± 10.0	186.9 ± 8.4	207.1 ± 7.8	224.6 ± 8.6 ^#^	215.2 ± 8.6	217.1 ± 9.0	198.7 ± 7.6
Functional bladder capacity (mL)	342.5 ± 17.6	345.3 ± 23.5	330.5 ± 12.2	349.9 ± 12.1	375.5 ± 13.2 ^#^	350.2 ± 12.9	358.2 ± 14.6	348.7 ± 13.6
Daytime frequency (times)	12.18 ± 0.99	12.00 ± 1.01	11.85 ± 0.50	9.76 ± 0.37^+, ##^	9.48 ± 0.33 ^##^	9.65 ± 0.30 ^##^	8.89 ± 0.26 ^##^	9.46 ± 0.36 ^##^
Nocturia (times)	1.88 ± 0.43	1.57 ± 0.33	1.77 ± 0.17	1.35 ± 0.12	1.20 ± 0.12 ^##^	1.22 ± 0.11 ^#^	1.18 ± 0.11 ^#^	1.41 ± 0.14
Urgency (times)	3.07 ± 0.29	2.83 ± 0.27	3.16 ± 0.39	2.13 ± 0.33	1.96 ± 0.34 ^#^	1.90 ± 0.35 ^#^	1.68 ± 0.31 ^#^	1.10 ± 0.23 ^##^
Uroflowmetry data								
Voided urine volume (mL)	293.8 ± 36.3	309.1 ± 42.5	296.2 ± 16.7	357.5 ± 20.1 ^#^	373.9 ± 21.5 ^#^	342.4 ± 16.8	344.2 ± 15.3	340.7 ± 13.4
Maximum flow rate (Qmax) (mL/s)	27.87 ± 2.59	26.46 ± 2.93	25.61 ± 1.69	27.45 ± 1.60	30.51 ± 1.26 ^#^	30.73 ± 1.37 ^#^	33.60 ± 3.26 ^#^	32.74 ± 1.83 ^#^
Post voided residual (PVR) (mL)	43.88 ± 14.27	56.43 ± 12.06	49.63 ± 6.18	40.88 ± 5.39	38.19 ± 4.66	30.04 ± 3.90 ^#^	31.24 ± 3.12 ^#^	23.86 ± 2.66 ^#^
OABSS score (points)								
Total score	7.80 ± 0.68	6.20 ± 0.98	7.42 ± 0.41	4.50 ± 0.20 ^+,##^	3.75 ± 0.25 ^##^	3.23 ± 0.20 ^##^	3.38 ± 0.24^##^	3.50 ± 0.26^##^
Daytime frequency	1.18 ± 0.14	1.10 ± 0.11	1.22 ± 0.07	0.84 ± 0.04 ^+,##^	0.77 ± 0.05 ^##^	0.77 ± 0.05 ^##^	0.70 ± 0.06 ^##^	1.00 ± 0.09
Nocturia	2.36 ± 0.29	2.09 ± 0.33	2.30 ± 0.10	1.45 ± 0.11 ^+,##^	1.31 ± 0.12 ^##^	1.17 ± 0.11^##^	1.33 ± 0.11 ^##^	1.33 ± 0.11 ^##^
Urgency	2.82 ± 0.52	2.55 ± 0.48	2.86 ± 0.19	1.70 ± 0.15 ^+,##^	1.33 ± 0.16 ^##^	0.96 ± 0.13 ^##^	1.02 ± 0.14 ^##^	0.83 ± 0.13 ^##^
Urgency incontinence	2.00 ± 0.68	1.88 ± 0.67	1.94 ± 0.21	0.89 ± 0.14 ^+,##^	0.81 ± 0.15 ^##^	0.64 ± 0.11 ^##^	0.78 ± 0.13 ^##^	0.88 ± 0.16 ^#^

Note: SE, standard error; W, week; W4, once per week, 4-week of LiESWT; W8, once per week, 8-week of LiESWT; F1, 1-month follow-up; F3, 3-month follow-up; F6, 6-month follow-up; OABSS, Overactive Bladder Symptom Scores. Values are means ± SE. ^#^ *p* < 0.05; ^##^ *p* < 0.01 vs. W0. + *p* < 0.05 vs. sham group. N = 15 (Sham) and N = 50 (LiESWT).

## Data Availability

Not applicable.

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
