# Peer review of "Low Intensity Extracorporeal Shock Wave Therapy as a Potential Treatment for Overactive Bladder Syndrome"

_biology, 2021, doi:10.3390/biology10060540_

Round 1

Reviewer 1 Report

This is an interesting assessment of a novel treatment for OAB. While the study appears to be well designed, the statistical analysis should be reconsidered to take account of the number of variables you are testing, and the discussion should be improved.

You quote results to many t-tests with no noted correction for multiple comparisons for 7 primary outcome variables, 6 timepoints and 2 groups. This manuscript would be strengthened by using a more sophisticated statistical analysis for comparing the two groups and multiple timepoints. Perhaps consult a statistician.

Which statistical tests did you do to confirm no change in voided volume/bladder capacity etc over the follow up periods? Please clarify.

What do you consider a ‘meaningful’ decrease in each questionnaire score or symptom frequency measure? Was this threshold defined before analysis?

Was UI measured on bladder diary or just scored on OABSS?

What is the purpose of breaking down OABSS scores when you have actual measurements of frequency/nocturia/urgency? You have already compared total score and make no mention of controlling for multiple comparisons.

Table 3 is not easy for the reader to follow. Please consider representing the important factors graphically.

Paragraph 4 appears out of place and repeats previous information from the methods.

What are the contraindications/sideffects for lieswt? Abdominal fat limitations? Diabetes? Implants? Please mention.

Please improve your discussion. How were the settings chosen – would you make recommendations for change or dependent on body type? Is it delivered on full or empty bladder? Can the patient feel it? Was it well tolerated? Was the weekly treatment agreeable to the patient, considering the benefit? Did anyone become dry? Please discuss the results of your study, not just potential mechanisms of action.

Author Response

Dear Dr. Bacescu,

We wish to thank the Editorial Board for the review of our manuscript entitled "Low intensity extracorporeal shock wave therapy as a potential treatment for overactive bladder syndrome", which is being considered by the Biology for publication.

By addressing every comment made by the reviewers, we have revised our manuscript. All the changes made in the manuscript are marked in red font.

We would like to thank you and the Editorial Board for the consideration and the intelligent review of our manuscript, which results in the revised manuscript of better quality.

Sincerely yours,

Yung-Shun Juan MD, PhD.

May 26th, 2021

Review 1:

Comments and Suggestions for Authors

This is an interesting assessment of a novel treatment for OAB. While the study appears to be well designed, the statistical analysis should be reconsidered to take account of the number of variables you are testing, and the discussion should be improved.

  • You quote results to many t-tests with no noted correction for multiple comparisons for 7 primary outcome variables, 6 time points and 2 groups. This manuscript would be strengthened by using a more sophisticated statistical analysis for comparing the two groups and multiple time points. Perhaps consult a statistician.

Response: Thanks for your professional recommendation. The current study was a prospective, randomized, single-blinded, controlled clinical trial to investigate the therapeutic effect of low intensity extracorporeal shock wave therapy (LiESWT) on participants with overactive bladder (OAB). Owing to some participants in the sham group who felt poor therapeutic effect and dropped out at the fifth or the sixth course, the data in the sham group only consisted of W4 data and were lack of W8, F1, F3, F6. Consequently, two quantitative statistical methods were analyzed. We painted illustration an illustration as shown in the below figure.

As suggested by the reviewer, the following information has been added to the Materials and Methods section on page 5, lines 157-163, as “In order to investigate the therapeutic effect of LiESWT on OAB, the scores of pre- and post-treatment (W0 vs. W4, W0 vs W8, W0 vs F1, W0 vs F3, W0 vs F6) for intragroup of patients were compared. The paired t-test was performed. On the other hand, the intergroup relationship (sham group vs. LiESWT group) of W0 and W4 data were evaluated by using the student's t-test. Moreover, the paired t-test was used to perform the measurement analysis for the intragroup before/after treatment and to calculate p-values for comparison.”

  • Which statistical tests did you do to confirm no change in voided volume/bladder capacity etc over the follow up periods? Please clarify.

Response: The paired t-test was used to clarify the effect of LiESWT on OAB, we compared the pre- and post-treatment (including follow up periods) scores (W4 vs. W0, W8 vs W0, F1 vs W0, F3 vs W0, F6 vs W0) for intragroup of patients. We took “voided urine volume” by uroflowmetry data as an example to clarify.

For example,

  • What do you consider a ‘meaningful’ decrease in each questionnaire score or symptom frequency measure? Was this threshold defined before analysis?

Response: Increased daytime frequency is the complaint by the patient who considers that he/she voids too often by day. There is no minimum voiding frequency serving as a threshold for the symptom, since it is highly subjective, and there is a wide overlap between normal and symptomatic. Paired t-test was used to perform the measurement analysis for the intragroup before/after treatment and to calculate p-values for comparison. We consider a ‘meaningful’ decrease in each questionnaire score or symptom frequency measured from statistically significant effects and patient feedback. We took “OABSS score” data as an example to illustrate a ‘meaningful’ decrease.

  • Was UI measured on bladder diary or just scored on OABSS?

Response: To investigate whether LiESWT improved the symptoms of OAB, including daytime frequency, nocturia, urgency and urgency incontinence (UI), we used OABSS scores to evaluate UI and showed results in Table 3, Figure 3b and Figure 3d. Meanwhile, 3-day bladder diary was used to investigate the daily times of frequency, nocturia, and urgency.

  • What is the purpose of breaking down OABSS scores when you have actual measurements of frequency/nocturia/urgency? You have already compared total score and make no mention of controlling for multiple comparisons.

Response: According to the International Continence Society (ICS) definition, OAB consists of daytime frequency, nocturia, urinary urgency with or without urgency incontinence. OAB is a symptom-based condition without physiological markers for disease activity. Bladder diary can actually determine daily times of frequency/nocturia/urgency. However, subjectively evaluate patient perception of bladder condition and OAB bothersome on quality of life is also important. OABSS is a single symptom score that employs a self report questionnaire to quantify OAB symptoms (shown below). Traditionally, a questionnaire has many items with the same minimum and maximum score. In contrast, with the OABSS, the scales vary. Since urgency is the core symptom of OAB, the design of OABSS is aimed at showing a clear separation between those with OAB and controls. In addition to OABSS total scores, breaking down the four sub-scores of OABSS and carefully interpret the difference between groups can help understand the actual effect of LiESWT.

  • Table 3 is not easy for the reader to follow. Please consider representing the important factors graphically.

Response: Thank you for your suggestion. In order to better understand and easy for the reader to follow Table 3, we presented the important factors graphically in Figures 2, 3 and 4.

  • Paragraph 4 appears out of place and repeats previous information from the methods.

Response: As suggested by the reviewer, we have revised the paragraph 3.4 in Result section. (please refer to page 10, lines 269-274)

  • What are the contraindications/side effects for LiESWT? Abdominal fat limitations? Diabetes? Implants? Please mention.

Response: No participant during whole timetable course complained adverse side effect for LiESWT, such as intolerable pain, hematuria or skin ecchymosis. Although LiESWT has little adverse effect in general practice, severe coagulopathy and malignant tumor at treatment sites and fetus in the treatment area were considered as absolute contraindications. Considering participant safety and interference factors, we exclude patient with severe coagulopathy, urologic cancer, and pregnant women in our study (please refer to Table 1). Abdominal fat distribution, including subcutaneous fat and visceral fat may influence the depth of the treatment site. Fortunately, most participants in the study are thin and have BMI between 18.6-26, abdominal fat accumulation has little effect on the present study. We also measured metabolic syndrome parameters in the study, hypercholesterolemia is common between sham and treatment groups.

  • Please improve your discussion. How were the settings chosen – would you make recommendations for change or dependent on body type? Is it delivered on full or empty bladder? Can the patient feel it? Was it well tolerated? Was the weekly treatment agreeable to the patient, considering the benefit? Did anyone become dry? Please discuss the results of your study, not just potential mechanisms of action.

Response: Thank you for your suggestion. We have added the fifth paragraph in Discussion section to improve our discussion (please refer to page 12, lines 335-353), as “Considering participant safety and interference factors, we have excluded patient with severe coagulopathy, urologic cancer, previous pelvic radiation and significant bladder outlet obstruction. Although LiESWT has little adverse effect in general practice, severe coagulopathy and malignant tumor at treatment sites were still considered as absolute contraindications. We also excluded pregnant women because fetus in the treatment area is another contraindication. Abdominal fat distribution, including subcutaneous fat and visceral fat may influence the depth of the treatment site. Fortunately, most participants in the study are thin and have BMI between 18.6-26, abdominal fat accumulation has little effect on the present study. In order to improve the effectiveness of LiESWT, the probe (applicator) was placed on the patient's lower abdomen around two fingers above the pubic symphysis. The probe was tilted to 45∘to target the bladder dome and bilateral bladder walls. Therefore, the effect of abdominal fat thickness will be decrease to the least level. The usage of ultrasound transmission gel over the abdominal skin and applicator to avoid air interference is also very important. All participants were asked to drink water, then receiving LiESWT treatment after filling the bladder to half of the functional bladder capacity by using bladder scan sonography. During LiESWT treatment, some participants felt a slight tingling sensation while some had no sensation. Overall, there were no adverse effects, such as intolerable pain, gross hematuria or skin ecchymosis occurred to the participants during the shock wave treatment.”

Reviewer 2 Report

This is an interesting study that provides meaningful results. The following are some comments.

  1. Is there any evidence or preclinical study to determine the energy of LiESWT in this study?
  2. Why the number different in of study group and sham group?
  3. Can patients receiving identical sham LiESWT treatment feel the pulse of the treatment?
  4. How to determine the target number of patients needed to get a meaningful result before the study?
  5. Because the bladder capacity different with different urine amount, did these patients receiving treatment with similar urine amount during treatment?

Author Response

Dear Dr. Bacescu and Dr. Reviewer 2,

We wish to thank the Editorial Board for the review of our manuscript entitled "Low intensity extracorporeal shock wave therapy as a potential treatment for overactive bladder syndrome", which is being considered by the Biology for publication.

By addressing every comment made by the reviewers, we have revised our manuscript. All the changes made in the manuscript are marked in red font.

We would like to thank you and the Editorial Board for the consideration and the intelligent review of our manuscript, which results in the revised manuscript of better quality.

Sincerely yours,

Yung-Shun Juan MD, PhD.

May 26th, 2021

Review 2:

Comments and Suggestions for Authors

This is an interesting study that provides meaningful results. The following are some comments.

  • Is there any evidence or preclinical study to determine the energy of LiESWT in this study?

Response: According to previous LiESWT treatment used for urologic disease, such as chronic pelvic pain syndrome and erectile dysfunction, LiESWT with 0.25 mJ/mm2 of energy, 3000 pulses and 3 pulses/second of frequency weekly was considered as a safe and therapeutic energy level. We add a paragraph in the “Introduction section” as “low intensity extracorporeal shock wave therapy (LiESWT) has been used widely for various types of urological diseases. For example, LiESWT treatment (0.10-0.25 mJ/mm2 and 3000-6000 pulses for 4-8 weeks) increased penile hemodynamics and induced penile tissue regeneration in erectile dysfunction (ED) patients. [13-15] LiESWT (0.10 to 0.25mJ/mm2, 3000 impulses for 4 weeks) was found to significantly improved pain, micturition, erectile function and QoL in chronic pelvic pain syndrome (CPPS) patients. It might be used as a potential therapy in treating non-bacterial prostatitis/CPPS [16]. We had found the pleasant result for severe CPPS patients who were refractory to many kinds of medical therapy, the pain scale and the severity of urinary symptoms were significantly improved after the LiESWT [17]. Additionally, our previous study suggested LiESWT attenuating SUI symptoms, diminishing bladder leaks and improving QoL after 8-week LiESWT (0.25 mJ/mm2 and 3000 pulses) [18] ” (please refer to page 2, lines 83-95)

  • Why the number different in of study group and sham group?

Response: Because many participants in the sham group withdraw from the study due to poor response and poor compliance, we cannot have same number participants in the study and sham groups. We have added the descriptions in the Discussion section (please refer to page 13, lines 354-358), as” First, some participants of the sham group withdraw at the fifth or the sixth course because there was no obvious improvement after sham treatment without energy transmission. The data in the sham group only W4 data and lack of W8, F1, F3 and F6. Therefore, the number of participants in the sham group (15 subjects) was different from the LiESWT group (50 subjects). “

  • Can patients receiving identical sham LiESWT treatment feel the pulse of the treatment?

Response: The sham group use air pad to block energy transmission during treatment, but the machine still emitted a shock wave generation. However, most participants still felt the pulsive vibration of the probe. We have added the related descriptions as “The probe of sham (placebo) group used air pad to block energy transmission, but the machine still emitted shock wave generation.” in the Materials and Methods section (please refer to page 4, lines 133-135). The air pad used was shown in the below figure.

  • How to determine the target number of patients needed to get a meaningful result before the study?

Response: The current study was a prospective, randomized, single-blinded, controlled clinical trial to investigate the therapeutic effect and duration of LiESWT on OAB symptoms. Sample size was estimated by the results of the pilot study. Therefore, we determined the target number of patients is 50 to get a meaningful result before the study.

  • Because the bladder capacity different with different urine amount, did these patients receiving treatment with similar urine amount during treatment?

Response: The treatment and sham groups had similar water intake and urine output during whole treatment and follow up periods (Table 3).

Reviewer 3 Report

Symptoms of OAB are suggestive of underlying detrusor overactivity. Overactivity of the detrusor muscle—neurogenic, myogenic, or idiopathic in origin—may result in urinary urgency and urgency incontinence

The authors in their previous publication (Low-Intensity Extracorporeal Shock Wave Therapy Ameliorates the Overactive Bladder: A Prospective Pilot Study. Biomed Res Int 2020; 2020: 9175676; published online 2020 Jul 6; doi: 10.1155/2020/9175676), and in this study, hypothesized that LiESWT could ameliorate detrusor overactivity by attenuating the inflammatory responses of the bladder, which consequently could alleviate OAB symptoms. Li-ESWT has been suggested to induce tissue angiogenesis, neuro-regeneration, anti-inflammation, and stem cell activation and recruitment as therapeutic mechanisms. I would suggest that the authors should be more specific and explain the basis of their hypothesis that LiESWT improves the overactive bladder symptoms including the basic science research published so far on the mechanism of action of LiESWT.

Author Response

Dear Dr. Bacescu and Dr. Reviewer 3,

We wish to thank the Editorial Board for the review of our manuscript entitled "Low intensity extracorporeal shock wave therapy as a potential treatment for overactive bladder syndrome", which is being considered by the Biology for publication.

By addressing every comment made by the reviewers, we have revised our manuscript. All the changes made in the manuscript are marked in red font.

We would like to thank you and the Editorial Board for the consideration and the intelligent review of our manuscript, which results in the revised manuscript of better quality.

Sincerely yours,

Yung-Shun Juan MD, PhD.

May 26th, 2021

Review 3:

Comments and Suggestions for Authors

Symptoms of OAB are suggestive of underlying detrusor overactivity. Overactivity of the detrusor muscle—neurogenic, myogenic, or idiopathic in origin—may result in urinary urgency and urgency incontinence

  • The authors in their previous publication (Low-Intensity Extracorporeal Shock Wave Therapy Ameliorates the Overactive Bladder: A Prospective Pilot Study. Biomed Res Int 2020; 2020: 9175676; published online 2020 Jul 6; doi: 10.1155/2020/9175676), and in this study, hypothesized that LiESWT could ameliorate detrusor overactivity by attenuating the inflammatory responses of the bladder, which consequently could alleviate OAB symptoms. Li-ESWT has been suggested to induce tissue angiogenesis, neuro-regeneration, anti-inflammation, and stem cell activation and recruitment as therapeutic mechanisms. I would suggest that the authors should be more specific and explain the basis of their hypothesis that LiESWT improves the overactive bladder symptoms including the basic science research published so far on the mechanism of action of LiESWT.

Response: Thanks for your professional recommendation. We have added the possible cellular signaling pathways for bladder overactivity modulated by LiESWT to the fourth paragraph in Discussion section (please refer to page 12, lines 325-334), as “The molecular mechanism of LiESWT activated protein kinase RNA-like ER kinase [31] pathway by increasing the phosphorylation levels of PERK and eukaryotic initiation factor 2a (eIF2α) and intensifying activating transcription factor 4 (ATF4) to increase the myotube formation in rat myoblast cells [32]. Zhu et al showed the effect of the combination of mesenchymal stem cells (MSC) and LiESWT on PI3K/AKT/ mTOR signal pathway and activated NO/cGMP signal pathway which induced vasodilatation and angiogenesis [33]. Additionally, LiESWT has protective effects on inflammation through lowering the expression of NGF, IL-6, IL-12, TNF-α, COX-2 and iNOS [28, 34]. LiESWT could en-hance endothelial NO synthase(eNOS) activity which result into the suppression of NF-κB to decrease inflammatory [35].”

We also added reference:

  1. Bodurka, A. Waters, Social-cognitive theory predictors of exercise behavior in endometrial cancer survivors, Health Psychol. 2013, 32, 1137-48, doi:10.1037/a0031712
  2. B. Wang, J. Zhou, L. Banie, A.B. Reed-Maldonado, H. Ning, Z. Lu, Y. Ruan, T. Zhou, H.S. Wang, B.S. Oh, G. Wang, S.L. Qi, G. Lin, T.F. Lue, Low-intensity extracorporeal shock wave therapy promotes myogenesis through PERK/ATF4 pathway, Neurourol Urodyn. 2018, 37, 699-707, doi:10.1002/nau.23380
  3. G.Q. Zhu, S.H. Jeon, W.J. Bae, S.W. Choi, H.C. Jeong, K.S. Kim, S.J. Kim, H.J. Cho, U.S. Ha, S.H. Hong, J.Y. Lee, E.B. Kwon, S.W. Kim, Efficient Promotion of Autophagy and Angiogenesis Using Mesenchymal Stem Cell Therapy Enhanced by the Low-Energy Shock Waves in the Treatment of Erectile Dysfunction, Stem Cells Int. 2018, 2018, 1302672, doi:10.1155/2018/1302672
  4. H.J. Wang, W.C. Lee, P. Tyagi, C.C. Huang, Y.C. Chuang, Effects of low energy shock wave therapy on inflammatory moleculars, bladder pain, and bladder function in a rat cystitis model, Neurourol Urodyn. 2017, 36, 1440-1447, doi:10.1002/nau.23141
  5. S. Mariotto, A.C. de Prati, E. Cavalieri, E. Amelio, E. Marlinghaus, H. Suzuki, Extracorporeal shock wave therapy in inflammatory diseases: molecular mechanism that triggers anti-inflammatory action, Curr Med Chem. 2009, 16, 2366-72, doi:10.2174/092986709788682119

Round 2

Reviewer 1 Report

Thank you for the improvement of the written portion of your study. The statistical analysis still needs some attention to support our conclusions.

Your use of t-tests without any correction for multiple comparisons is inappropriate. You appear to have performed 27 paired t-tests (per your diagram) on 14 different parameters (per the table). This increases the possibility of a type I error, and the use of a p<0.05 threshold is therefore not appropriate. At the very least, a correction for multiple comparisons (e.g. Bonferroni) should be used, but likely, a better statistical approach should be taken.

Re: a ‘meaningful change’ in symptoms score: a statistically significant difference is not equivalent to a meaningful clinical change or meaningful improvement in patient bother. Even small changes in scores can be statistically significant, but do not translate to clinically meaningful improvement. For instance, literature suggests the minimal clinically meaningful change in OABSS is -3. What is the case with the other measures?

Author Response

Dear Dr. Bacescu and Dr. Review 1,

We wish to thank the Editorial Board for the review of our manuscript entitled "Low intensity extracorporeal shock wave therapy as a potential treatment for overactive bladder syndrome", which is being considered by the Biology for publication.

By addressing every comment made by the reviewers, we have revised our manuscript. All the changes made in the manuscript are marked in red font.

We would like to thank you and the Editorial Board for the consideration and the intelligent review of our manuscript, which results in the revised manuscript of better quality.

Sincerely yours,

Yung-Shun Juan MD, PhD.

June 5th, 2021

Review 1:

Comments and Suggestions for Authors

Thank you for the improvement of the written portion of your study. The statistical analysis still needs some attention to support our conclusions.

  • Your use of t-tests without any correction for multiple comparisons is inappropriate. You appear to have performed 27 paired t-tests (per your diagram) on 14 different parameters (per the table). This increases the possibility of a type I error, and the use of a p<0.05 threshold is therefore not appropriate. At the very least, a correction for multiple comparisons (e.g. Bonferroni) should be used, but likely, a better statistical approach should be taken.

Response: As suggested by the review, we have compared Student's t test and paired t-test with one-way ANOVA following post-hoc Tukey’s test as below. We also modified our Figure 2, Figure 3, and Table 3. The following information has been added to the Materials and Methods section on page 5, lines 155-166, as “Student t-test was performed for the intergroup comparison. The paired t-test and one-way analysis of variance were used to perform repeated measurement analyses for intragroup before/after treatment [21]. In order to clarify the therapeutic effect of LiESWT on OAB, the scores of pre- and post-treatment for intragroup of patients were compared. The paired t-test was performed in the sham group (W0 vs. W4). The post-hoc Tukey’s honestly significant difference tests were used to make comparison between the LiESWT subgroups and to calculate p-values for comparison [21].” We also modified our Result section (lines 185 and 188 on page 6; line 189 on page 7; lines 209, 214-216, and 218-219 on page 8; line 237 on page 9) and Figure legend section (Figure 2, line 200 on page 6).   

Reference :

  1. Tervonen, T., Karjalainen, K., Periodontal disease related to diabetic status. A pilot study of the response to periodontal therapy in type 1 diabetes, J Clin Periodontol. 1997, 24, 505-10, doi:10.1111/j.1600-051x.1997.tb00219.x.

Fig 2b and 2c

  1. Student’s t-test, Paired t-test and One-way ANOVA

Fig 3a and 3c

  1. Student’s t-test, Paired t-test and One-way ANOVA

Fig 3b and 3d

  1. Student’s t-test, Paired t-test and One-way ANOVA

Table 3.

  1. Student’s t-test, Paired t-test and One-way ANOVA

Re: a ‘meaningful change’ in symptoms score: a statistically significant difference is not equivalent to a meaningful clinical change or meaningful improvement in patient bother. Even small changes in scores can be statistically significant, but do not translate to clinically meaningful improvement. For instance, literature suggests the minimal clinically meaningful change in OABSS is -3. What is the case with the other measures?

Response: Thanks for your professional recommendation. As mentioned, a statistically significant difference is not equivalent to a meaningful clinical change or meaningful improvement in patient bother. However, Gotoh et al. had revealed a linear tendency between the changes in the OABSS and symptom improvement. They also shown a change -3 in OABSS total score is the minimal threshold for a clinical meaningful change [37]. In the present study (Table 3), in the sham group, the OABSS scores only decrease from the baseline W0 (7.80 ± 0.68) to W4 (6.20 ± 0.98). However, in the treatment group, the OABSS scores decrease from the baseline W0 (7.42 ± 0.41) to W8 (3.75 ± 0.25), F1 (3.23 ± 0.20), F3 (3.38 ± 0.24), and F6 (3.50 ± 0.26), respectively. The difference in OABSS between pre-treatment and post-treatment is greater than 3, showed that LiESWT not only had statistically significant difference but also provided clinically meaningful improvement for these OAB patients. We had added this statement in the Discussion (please refer to lines 359-367 on page 12 and lines 368-369 on page 13) and add the reference in the manuscript.

Reference list:

  1. Gotoh, M., Homma, Y., Yokoyama, O., Nishizawa, O., Responsiveness and minimal clinically important change in overactive bladder symptom score, Urology. 2011, 78, 768-73, doi:10.1016/j.urology.2011.06.020.

Reviewer 2 Report

The authors could answer all the questions and revised their manuscript.

Author Response

Dear Dr. Bacescu and Dr. Reviewer 2,

We wish to thank the Editorial Board for the review of our manuscript entitled "Low intensity extracorporeal shock wave therapy as a potential treatment for overactive bladder syndrome", which is being considered by the Biology for publication.

By addressing every comment made by the reviewers, we have revised our manuscript. All the changes made in the manuscript are marked in red font.

We would like to thank you and the Editorial Board for the consideration and the intelligent review of our manuscript, which results in the revised manuscript of better quality.

Sincerely yours,

Yung-Shun Juan MD, PhD.

June 5th, 2021

Review 2:

Comments and Suggestions for Authors

The authors could answer all the questions and revised their manuscript.

Response: Thanks for your professional recommendation.

Reviewer 3 Report

Overall, very interesting addition to the armamentarium to treat the common condition of overactive bladder. Randomized studies utilizing larger patient population are warranted to corroborate the findings of the authors.

Author Response

Dear Dr. Bacescu and Dr. Reviewer 3,

We wish to thank the Editorial Board for the review of our manuscript entitled "Low intensity extracorporeal shock wave therapy as a potential treatment for overactive bladder syndrome", which is being considered by the Biology for publication.

By addressing every comment made by the reviewers, we have revised our manuscript. All the changes made in the manuscript are marked in red font.

We would like to thank you and the Editorial Board for the consideration and the intelligent review of our manuscript, which results in the revised manuscript of better quality.

Sincerely yours,

Yung-Shun Juan MD, PhD.

June 5th, 2021

Review 3:

Comments and Suggestions for Authors

Overall, very interesting addition to the armamentarium to treat the common condition of overactive bladder. Randomized studies utilizing larger patient population are warranted to corroborate the findings of the authors.

Response: Thanks for your professional recommendation.

Round 3

Reviewer 1 Report

Thank you for carefully revising the stats in this paper.